



**An iterative algorithm to simultaneously retrieve aerosol extinction and effective radius profiles using**
**the CALIOP lidar**
Liang Chang[1], Jing Li[1, #], Jingjing Ren[2], Changrui Xiong[1], Lu Zhang[3,4]
*[1] Department of Atmospheric and Oceanic Sciences, School of Physics, Peking University, Beijing 100871,*
*China*
*[2] Intelligent Science & Technology Academy Limited of CASIC*
*[3] Key Laboratory of Radiometric Calibration and Validation for Environmental Satellites, National Satellite*
*Meteorological Center (National Center for Space Weather), China Meteorological Administration, Beijing*
*100081, China*
*[4] Innovation Center for FengYun Meteorological Satellite (FYSIC), Beijing 100081, China*
*# Correspondence to*: Jing Li (jing-li@pku.edu.cn)

12                                                    **Abstract**

The Cloud-Aerosol Lidar with Orthogonal Polarization (CALIOP) onboard the Cloud-Aerosol Lidar and
Infrared Pathfinder Satellite Observation (CALIPSO) satellite has been widely used in climate and
environment studies to obtain the vertical profiles of atmospheric aerosols. To retrieve the vertical profile of
aerosol extinction, the CALIOP algorithm assumes column-averaged lidar ratios based on a clustering of
aerosol optical properties measured at surface stations. On one hand, these lidar ratio assumptions may not
be appropriate or representative at certain locations. One the other hand, the two-wavelength design of
CALIOP has the potential to constrain aerosol size information, which has not been considered in the
operational algorithm. In this study, we present a modified inversion algorithm to simultaneously retrieve
aerosol extinction and effective radius profiles using two-wavelength elastic lidars such as the CALIOP.
Specifically, a look-up table is built to relate the lidar ratio with the Ångström exponent calculated using
aerosol extinction at the two wavelengths, and the lidar ratio is then determined iteratively without a priori



assumption. The retrieved two-wavelength extinction at each layer is then converted to particle effective
radius assuming a lognormal distribution. The algorithm is tested on synthetic data, Raman lidar
measurements and then finally the real CALIOP backscatter measurements. Results show improvements over
the CALIPSO operational algorithm by comparing with ground-based Raman lidar profiles.
**1 Introduction**
Atmospheric aerosols have important impacts on the physical and chemical processes in atmosphere, as well
as the climate system and public health. Optical properties of aerosols are critical in quantifying their radiative
effects in the Earth's climate system. Moreover, the vertical distribution of aerosol properties, such as its
extinction coefficient and particle size, is one of the key elements to assess climate effect (Ipcc, 2023). Direct
aerosol radiative forcing, which plays an important role in the Earth's energy budget, is impacted by the
vertical distribution of aerosols, especially that for absorbing aerosols (Goto et al., 2011; Eswaran et al., 2019;
Zhang et al., 2022). The vertical profiles of aerosol optical properties is also essential estimating the solar
heating rate (Kudo et al., 2016), and establishment of aerosol parameterization schemes for satellite remote
sensing (He et al., 2016). Although its importance is widely recognized, aerosol vertical distribution is very
difficult to monitor globally. Lidar is a major technique for obtaining the profiles of the aerosol properties,
which has been used in ground-based and satellite remote sensing systems. Especially, spaceborne lidar is an
effective way to observe the global distribution of aerosols. The Cloud-Aerosol Lidar with Orthogonal
Polarization (CALIOP) on the CALIPSO (The Cloud-Aerosol Lidar and Infrared Pathfinder Satellite
Observation) satellite, the only long-term orbiting spaceborne lidar to date, was launched on 28 April 2006.
The CALIOP is a three-channel Mie-scattering lidar system, which contains two wavelengths of $532nm$
(perpendicular & parallel polarization channel) and $1064nm$. It is the first polarization lidar to provide three-
channel elastic backscatter signals of global atmospheric measurements. The official aerosol retrieval
algorithm of CALIOP involves three modules, namely the Selective Iterated BoundarY Locator (SIBYL),
the Scene Classification Algorithm (SCA), and the Hybrid Extinction Retrieval Algorithms (HERA). The





HERA algorithm requires a lidar ratio (extinction-to-backscatter ratio of aerosols), which is provided by the
SCA. The SCA uses three CALIOP channels ($532nm$ parallel, $532nm$ perpendicular and $1064nm$ channels)
to obtain the lidar ratio from the 6 groups of assumed column-averaged lidar ratios based on a clustering of
aerosol optical properties measured at surface stations (Winker et al., 2009). However, due to the limited
coverage and spatial representativeness of surface stations, these lidar ratio assumptions may not be
appropriate or representative at certain locations (Josset et al., 2011), which is an important source of retrieval
uncertainty.

The lidar ratio is dependent on the chemical composition, shape, particle size distribution of aerosols,

as well as the lidar wavelength (Burton et al., 2012), which is a critical parameter required for solving the
Mie-scattering lidar equation using the Klett (Klett, 1985) or Fernald (Fernald, 1984) methods. Previous
studies have developed algorithms to determine the lidar ratio iteratively for two-wavelength Mie scattering
lidars. Potter (1987) first introduced the two-wavelength lidar inversion technique to retrieve the aerosol
transmission with a constant lidar ratio in two independent wavelengths. Ackermann (Ackermann, 1997,
1998) developed an iterative method to obtain the variable lidar ratio from two-component (i.e., molecule
and aerosol) atmospheres by transcendental equation. Rajeev and Parameswaran (1998) proposed a new
method using the Mie theory calculated aerosol optical properties with Junge distribution of aerosols to
determine the lidar ratio by iteration. Lu et al. (2011) made an attempt to improve the two-wavelength lidar
inversion by iterative method, but failed to consider the size distribution of aerosols which may introduce
uncertainties in the inversion. Moreover, these studies mostly only gave the aerosol extinction profile without
retrieving the vertical distribution of aerosol size information. The algorithms were also mostly applied to
theoretical data or ground lidar measurements. The application to space lidars such as CALIOP is challenging
and thus limited.

In view of the above discussions, this study aims to provide a modified two-wavelength lidar

inversion algorithm to retrieve the vertical distribution of both aerosol extinction and particle effective radius,
avoiding the complex calculation confronted in the previous two-wavelength lidar inversion methods. The



algorithm is tested on synthetic data, surface Raman lidar and is finally applied to CALIOP measurements,
in order to better demonstrate its operational feasibility. The paper proceeds with descriptions of the inversion
algorithm in Sect. 2. Sect. 3 presents the application of the algorithm to the Raman lidar and CALIOP with
an analysis of retrieval uncertainties provided in Sect. 4. The study concludes in Sect. 5 with a brief discussion
in the context of relevant lidar algorithms.

## 2 Description of the lidar inversion algorithm

The modified inversion algorithm retrieves the profiles of aerosol extinction and effective radius at two
wavelengths, by solving the lidar equation using the Fernald method (Fernald, 1984) with a look-up table.

### 2.1 Solving the lidar equation

For each wavelength with a complete overlap between the fields of view of the laser and of the receiver, the
lidar equation with calibration and range-correction can be expressed as:
$$\beta'(R) = \frac{P(R)R^2}{E_0\xi} = \left[\beta_m(R) + \beta_p(R)\right]T_m^2(R)T_p^2(R),$$    (1)
where
$$T^2(R) = e^{-2\tau(R)},$$    (2)
$$\tau(R) = \int_{R_0}^{R} \sigma(r)dr,$$    (3)

In Eq. (1-3), $\beta'(R)$ is the attenuated backscatter coefficients (calibrated and range-corrected signal)

from distance $R$; $P(R)$ is the measured signal after background subtraction and artefact removal from
distance $R$; $E_0$ is the average laser energy for the single-shot; $\xi$ is the lidar system parameter; $\beta(R)$ and $\sigma(R)$
are the volume backscatter and extinction coefficient at range $R$, respectively; $T^2(R)$ is the one-way
transmittance from the lidar to the scattering volume at range $R$; $\tau(R)$ is the optical depth at range R; and the
subscripts $M$ and $P$ denote the portions of air molecules and aerosols, respectively.

In order to facilitate calculation, the transmittance of air molecules $T_m^2(R)$ is separated from $\beta'(R)$

to obtain the $E(R)$ as


$E(R) = \frac{\beta'(R)}{T_m^2(R)},$ (4)
As is well known, lidar back scatter signal is also subject to multiple scattering effects. These effects
are typically small for low to moderate aerosol loading, and is only significant for optically thick clouds [7].
Therefore, we neglect multiple scattering effects here and consider that the lidar ratio ($S(R)$) of aerosols is
range dependent in single-scatter approximation, which can be written as
$S(R) = \frac{\sigma_p(R)}{\beta_p(R)},$ (5)
In the following, we use the Fernald method (Ackermann, 1998) to obtain the aerosol extinction
coefficient at distance $R$ as
$\sigma_p(R) = S(R)\left\{E(R)e^{-2\int_{R_0}^R S(r)\beta_m(r)dr}\left[C - 2\int_{R_0}^R E(r)S(r)e^{-2\int_{R_0}^r S(r')\beta_m(r')dr'}dr\right]^{-1} - \beta_m(R)\right\},$ (6)
where
$C = \frac{\beta'(R_0)}{\beta_p(R_0)+\beta_m(R_0)},$ (7)
The backscatter and extinction coefficient of air molecules can be determined with the Rayleigh
scattering theory with the observed atmospheric profile (Bodhaine et al., 1999) as
$\sigma_m(R,\lambda) = \frac{C_s(\lambda)P(R)}{T(R)},$ (8)
$\beta_m(R,\lambda) = \frac{\sigma_m(R,\lambda)}{\frac{8\pi}{3}k_{b\omega}(\lambda)},$ (9)
Where $P(R)$ and $T(R)$ are the atmospheric pressure ($hPa$) and temperature ($K$) at distance $R$, respectively.
$C_s(\lambda)$ and $k_{b\omega}(\lambda)$ are the atmospheric molecular constant related to the wavelength $\lambda$. Hostetler et al. (2006)
suggested the values of $C_s(\lambda)$ and $k_{b\omega}(\lambda)$ at 532nm and 1064nm as $C_s(532nm) = 3.742 \times 10^{-6}(K/$
$hPa/m)$; $C_s(1064nm) = 2.265 \times 10^{-7}(K/hPa/m)$; $k_{b\omega}(532nm) = 1.0313$; $k_{b\omega}(1064nm) = 1.0302$.
Thus, the aerosol extinction coefficient profiles can be obtained by Eq. (6) with an unknown variable
of the lidar ratio. The two-wavelength lidar can give two independent profiles of attenuated backscatter





coefficients, from which the aerosol extinction coefficient profiles can be calculated by assuming the lidar
ratios at the two wavelengths.

For two wavelengths ($\lambda_1$ & $\lambda_2$), the Ångström exponent ($AE$) at distance $R$ is defined as:

$$AE(R) = -\frac{ln\left[\frac{\sigma_P(R,\ \lambda_1)}{\sigma_P(R,\ \lambda_2)}\right]}{ln\left[\frac{\lambda_1}{\lambda_2}\right]}, \tag{10}$$

Because $AE$ is related to particle effective radius, which is a primary factor determining the lidar ratio,

an $AE$-lidar ratio relationship can be established and used to determine the lidar ratio at each layer, which can
then be used to retrieve aerosol extinction profiles from two-wavelength lidar measurements.
**2.2 Look-up table**
By assuming spherical particles with some size distribution, the aerosol extinction coefficients and
backscatter coefficients can be calculated by Eq. (11-12):
$$\sigma_p(\lambda) = \int_{r_{min}}^{r_{max}} Q_e(\lambda, r)\,\pi r^2 n(r)dr, \tag{11}$$

$$\beta_p(\lambda) = \int_{r_{min}}^{r_{max}} Q_b(\lambda, r)\,\pi r^2 n(r)dr, \tag{12}$$

Where $n(r)$ represents the volume-size distribution of particles; $r_{max}$ and $r_{min}$ are the maximum and
minimum of the particle effective radius, respectively; $Q_e(\lambda, r)$ and $Q_b(\lambda, r)$ denote the extinction efficiency
and backscatter factor of the particle with size $r$ at wavelength $\lambda$. The size parameter is defined as $x \equiv 2\pi r /$
$\lambda$ , where $1 < x < 50$ for typical aerosols and thus the Mie scattering theory can be applied.

As the limited information provided by two-wavelength lidar, we assume the volume-size distribution

of aerosols conform to the lognormal distribution, and the size distribution is expressed as follows:
$$n(r) = \frac{N}{\sqrt{2\pi}s_d} e^{-\frac{(r-\bar{r})^2}{2s_d^2}}, \tag{13}$$

Where $N$ is the total particle concentration; $\bar{r}$ is the average particle radius; $s_d$ is the standard deviation.
When the $s_d$ is a constant in the same aerosol, the $AE$ can be determined by the $r$.

We choose the six types of aerosols with their parameters in Table 1, which is consistent with the

aerosol classification used in the operational algorithm of CALIOP. From Table 1, Type 3 denotes the



scattering aerosols, Type 2 shows both strong scattering and absorption, whereas other types are moderate
scattering or absorbing. Combining Eqs. (5, 10-13), the relationship between Ångström exponent ($AE$) and
lidar ratio ($S$), as well as that between $AE$ and particle effective radius ($r$) can be formulated as look-up tables
for different refractive indices, as shown in Figure 1. Note that in Figure 1, it is easy to determine $S_{532nm}$,
$S_{1064nm}$ and $\bar{r}$ by the unique $AE$ calculated from the lidar equation for a fixed aerosol type.

### 2.3 The iterative inversion procedure

After constructing the look-up table, we design the following iterative procedure to simultaneously retrieve
aerosol extinction and effective radius profiles. Firstly, we calculate the extinction coefficients ($\sigma_{532nm}$ &
$\sigma_{1064nm}$) of two wavelengths ($532nm$ & $1064nm$) from an initial guess of the lidar ratios ($S^0_{532nm}$ &
$S^0_{1064nm}$) by solving the lidar equation (Eq. 6), then obtain the Ångström exponent ($AE$) through Eq. (10).
Secondly, the look-up table are used to determine a set of new lidar ratios ($S'_{532nm}$ & $S'_{1064nm}$), which is used
to calculate the new $\sigma_{532nm}$ & $\sigma_{1064nm}$ and Ångström exponent ($AE'$). This procedure is repeated until the
difference between the updated $AE'$ and previous AE reduces to a very small value (e.g., $10^{-3}$). The final AE
is converted to effective radius from the AE-$\bar{r}$ look-up table, and the final values of $\sigma_{532nm}$, $\sigma_{1064nm}$, and $\bar{r}$
are the retrieved results of this layer. The above iterative algorithm is summarized into Figure 2.
Although in theory, our algorithm can retrieve aerosol extinction and effective radius at each layer,
in reality the measurement noise may cause the inversion of certain layers fail to converge. In these cases,
we assume that this layer has the same aerosol type and size distribution as its adjacent layer, and then these
two layers are combined into a new layer to continue with the inversion.

### 2.4 Test of the algorithm with synthetic data

For verifying the feasibility of the inversion algorithm, we first conduct some retrieval tests using synthetic
data from Mie scattering and radiative transfer simulations. We assume a hypothesized profile of effective
radius, backscatter and extinction coefficients of the aerosols, and use the American atmospheric model in
1976 (National Geophysical Data, 1992) for molecular scattering, and calculate the attenuated backscatter



profiles according to the lidar equation. We then apply our algorithm to retrieve the aerosol property profiles
from these simulated lidar signals and compare them with the initial assumptions.

To save space, we only present the results for the reflective aerosol model, and results for other aerosol

types are similar.  The simulated attenuated backscatter profiles for the two wavelengths are shown in Figure
3, and the results of our inversion and their comparison with the assumed profiles are shown in Figure 4. It
is clearly seen that the results of the inversion are in good agreement with the assumed profiles. The RRMSE
(Relative Root Mean Square Error) between retrieved and assumed profiles of extinction coefficient, average
particle effective radius and lidar ratio are all below 0.01%, which proves the validity of the algorithm in
theory. Note that typically, selection of aerosol type is critical as incorrect assumption of aerosol refractive
index will result in divergence of the algorithm and thus yield no valid retrieval.
**3 Application to real lidar measurements**
Before applying our algorithm to CALIOP measurements, we first use Raman lidar measurements to test its
accuracy as Raman lidars can directly retrieve aerosol extinction profiles without assuming a lidar ratio.
**3.1 Application to Raman lidar measurements**
A Raman lidar  (Model LR231-D300, Raymetrics S.A, Greece) is installed on top of an 8-floor building at
the Peking University site (39°59′N, 116°18′E, 53m above sea level). It can provide the extinction and
backscatter coefficient at $532nm$ by Raman inversion (Ansmann et al., 1990) without the need to assume the
lidar ratio. To test our inversion algorithm, we apply it to the elastic backscatter signals at 532 and 1064nm
and compare the retrieved extinction profile at 532nm with that retrieved with the Raman method.  but an
approximation of $AE$ is used in the inversion at $1064nm$. We applicate the modified inversion algorithm to
the cases of four different aerosol types. To facilitate the determination of the initial value, we use the mothed
of remodelling downward attenuated backscatter from ground-based lidar  (Tao et al., 2008) to reconstruct
the Raman lidar measurements at wavelength of $532nm$ and $1064nm$, which are showing Figure 5-8(a).





We examined four cases in December 2017, as shown in Figures 5-8. The cases on 2 and 21 December 2017
both indicate that the extinction coefficient decreases sharply with altitude, and the maximum values occur
near the ground (Figure 6b & 7b). The other two cases on December 1 and 23 respectively show the features
of an elevated aerosol layer with maximum extinction found above the surface. In all four cases, our retrieval
results (red curves) agree well with those retrieved by the Raman method, with RMSE lower than 0.05. The
lidar ratio profiles retrieved by our algorithm also agree well with obtained from Raman method in some
ranges, except these spikes at the highest or lowest point, may be caused by the uncertainty of boundary. The
aerosol particle effective radius slightly increases with altitude and the peak (corresponding to ~$0.1\mu m$)
appear at ~0.7km and ~1.7km on 1 and 23 December 2017 (Figure 5d & 8d), respectively. Similar results
were found by Zhang et al. (2009) and Cai et al. (2022) with aircraft measurements over Beijing and the
Loess Plateau in China respectively, which are mainly associated with long range aerosol transport. The
variability of particle effective radius profiles in Figure 6d is a typical feature for low (and stable) PBL
(Planetary Boundary Layer), which results in both particles and water vapor accumulating near PBL top and
thus remarkable hygroscopic growth of particle size may occur (Yang et al., 2020). The case for Dec 21
(Figure 7d) shows relatively large particle size below~1.4km but sharply decreases. This is likely related to
the domination of local pollutions and insignificant PBL temperature inversion (Li et al., 2022; Liu et al.,
2009; Zhang et al., 2009).

### 3.2 Application to CALIOP measurements

We further apply our algorithm to real CALIOP measurements. To test its performance, we collocate
CALIOP profiles with those from surface-based Raman lidar measurement within the European Aerosol
Research LIdar NETwork (EARLINET, www.earlinet.org, (Matthias et al., 2004). Aerosol profiles from the
Napoli (southern Italy, 40.838°N , 14.183°E , 118m above sea level), Evora (south-central
Portugal, 38.5678°N, −7.9115°E, 293m above sea level) and Warsaw (east-central, 52.21°N, 20.98°E,
112m above sea level) stations have the best match with CALIOP and high data quality in cloudless sky, are
primarily used to validate the retrieval results. The CALIPSO overpass times for the chosen cases and the




corresponding horizontal distances between the sub-satellite point and ground-based Raman lidar site are
listed in Table 2.

To compare with the lidar returns measured by CALIOP (down-looking) and ground-based Raman

lidar (up-looking), we still use the mothed of remodelling downward attenuated backscatter from ground-
based lidar (Tao et al., 2008) to reconstruct the downward attenuated backscatter signals for the ground-
based Raman lidar. The attenuated backscatter signals of CALIOP was averaged for 163 nearby sub-satellite point
profiles (CALIPSO ground track range of about 30km within 8s) (Lu et al., 2011; Wang et al., 2007), obtained
from CALIOP level 1B products, to improve the signal-to-noise ratio.

The attenuated backscatter profiles at $532nm$ from CALIOP agree well with those from the Napoli

Raman Lidar (NRL), as shown in Figures 9-14(a). The initial altitude of inversion (the upper boundary of the
aerosol layer) is determined by the variation of attenuated backscatter signal and volume linear depolarization
ratio at $532nm$. Comparison between our inversion results, CALIOP operational results and Raman results
is shown in Figure 9-14(c).

The CALIOP operational product only provides retrievals for three cases considered, namely 20

August 2006, 20 June 2007 and 22 July 2007. In all three cases, the aerosol extinction profiles of our
algorithm (red curve) appear in better consistency with Raman lidar results. Our algorithm successfully
corrects the overestimation for the August 20 2006 and July 22, 2007 cases. For the June 20, 2007 case, the
operational results show a lower peak at ~1.7km and a secondary peak at ~4km, both of which are absent in
the Raman profile, and our results agree well with Raman in both the shape and magnitude. In the other three
cases, CALIOP does not provide Level 2 retrieval results. Our algorithm is able to retrieve and the extinction
profiles agree well with Raman lidar observations. Our retrievals do show more fluctuations compared to
Raman lidar, possibly due to the noises in the attenuated backscatter profiles of CALIOP. Because Raman
lidar does not provide retrieval of aerosol effective radius profiles, we compare the lidar ratio profiles by our
algorithm and the Raman algorithm. Overall, our algorithm produces lidar ratios varying in a relatively small
range around 50, whereas Raman lidar ratios can vary from ~10 to 200. Also, the Raman lidar ratios tend to





change sharply at the highest or lowest point, which may be caused by the inversion errors at the boundary.
By removing these spikes, the differences of the lidar ratio between CALIOP and Raman is obviously reduced.
In general, the aerosol particle effective radius increases with altitude, similar to Figures 5d and 8d, but the
fluctuations of the profiles may also be caused the noise in the CALIOP measurement.
When examining the CALIOP backscatter measurements, we found that the backscatter signal at
1064nm is often stronger than that at 532nm after 2010, which is unphysical and possibly due to issues such
as calibration and lidar degradation. As a result, the remodeled backscatter profiles of CALIOP appear noisier
and do not exactly match those from Raman lidar for the Evora and Warsaw stations, which only have
collocated measurements in 2019 and 2020 (Figure 15-19a). Our retrieved extinction profiles also agree
reasonably well with those by Raman lidar (Figure 15-19b), with the lidar ratio profiles and aerosol particle
effective radius profiles similar to the cases at Naples. By contrast, the extinction profiles of the official
CALIPSO product show large deviations from the Raman profile with unphysical spikes (Figure 16b),
incomplete profiles (Figure 17&18b) or no retrievals (Figure 15b).
**4 Uncertainty analysis**
Uncertainties in aerosol extinction and effective radius profiles from our two-wavelength inversion algorithm
are mainly due to the measurement noise (e.g., the signal statistical error, the estimations of molecular optical
properties, etc.), calibration errors, assumption errors (e.g., single-scatter approximation) and the look-up
table. In this section, we mainly analyze the errors associated with the look-up table.
Since the value of $AE$, which is the key variable in the iterative process, is obtained from the look-up
table, the errors on the hypothesis of aerosol refractive index, size distribution and shape in each aerosol layer
will affect the variability of lidar ratio in solving the lidar equation. Figure 20 shows the relationship between
spherical aerosol particle radius and $AE$ in different aerosol refractive indices. For aerosol particles with the
same size, the real part of the refractive index ($m_r$) mainly affects the cycle period of $AE$, and the imaginary
part ($m_i$) directly impacts its range of variability. In addition, AE is not quite sensitive to coarse particles,




which limit the applicability of our algorithm primarily to fine mode aerosols. The spherical assumption also
adds to the uncertainty in the existence of non-spherical particles, such as dust.
Although the significant difference of these six aerosol types in look-up table can ensure the reasonable
inversion result come from a specific aerosol type, the limited look-up table also restrict the inversion of
other aerosol types. As the different type of aerosols in the aerosols optical parameters database of CEOS-
Chem (http://wiki.seas.harvard.edu/geos-chem) show that the relative change of complex imaginary parts of
refractive index is greater than its real parts (e.g. at $532nm$: $1.3 < m_r < 1.7$ & $0 < m_i < 0.4$), which tells
our look-up table need to pay more attention to the complex imaginary parts of refractive index in the future.
**5 Summary and discussion**
In this study, we described a modified lidar inversion algorithm to retrieve aerosol extinction and size
distribution simultaneously from two wavelengths elastic lidar measurements. Its major advantage over the
operational CALIOP algorithm is that the lidar ratio of each layer is determined iteratively by the lidar ratio-
AE look-up table.  The algorithm was applied to the ground-based Raman lidar measurements at the PKU
site, as well as to CALIOP measurements. The comparison results indicate that the retrieved aerosol
extinction coefficient profiles by our method using CALIOP attenuated backscatter measurements are in
good agreement with Raman lidar measurements. Characteristics of aerosol effective radius profiles are also
retrieved, which can be used as a reference for aerosols size information.
In comparison with the iterative method by transcendental equation (Ackermann, 1997, 1998), our
inversion uses the look-up table to simplify the complex calculation. Cao et al. (2019) develop a lidar-ratio
iteration method to invert the particle-size distribution with assumed Junge distribution, but the method was
just used in simple simulation without actual tests. Although Lu et al. (2011) invert the aerosol backscatter
coefficient profiles from CALIPSO lidar measurements by iterative method, failed to consider the size
distribution of aerosols which may introduce uncertainties in the inversion. Compared with other modified
CALIOP inversions by combining other measurements, such as ground-based lidar (Wang et al., 2007), our
inversion is weaker by the space-time limitations.
However, this study still bears certain limitations. The current algorithm is primarily suitable for fine
mode spherical particles, such as urban pollution, and considers the change of aerosol size (thus lidar ratio)
with altitude, due to long range transport, vertical mixing, hygroscopic growth, etc. Non-spherical particles
such as dust will be explored in the next step, possible by taking advantage of the depolarization ratio
measurement that is not used here. Another drawback is that although the algorithm does not need to assume
a lidar ratio, the complex refractive index still needs to be assumed. As discussed above, the lidar ratio is
very sensitive to the imaginary part and an incorrect assumption may induce errors or even makes the
algorithm unable to converge. Therefore, this algorithm is mostly suitable when there is no significant change
in aerosol type vertically. Finally, the polarization channel of CALIOP may contain additional aerosol type
information but is only used when determining the initial refractive index (excluding dust) here. We also plan
to refine our look-up table by incorporating polarization in order to improve the accuracy of the retrieval.
**Data availability**
All raw data can be provided by the corresponding authors upon request.
**Author contributions**
LC and JL planned the research; LC, JL, JR, CX, and LZ developed the algorithm; LC and JL analyzed the
results; LC and JL wrote the manuscript.
**Competing interests**
The authors declare that they have no conflict of interest.



## Acknowledgement

This study is funded by National Natural Science Foundation of China (NSFC) Grant No. 42175144.



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

Z.: Spatial Distribution and Impacts of Aerosols on Clouds Under Meiyu Frontal Weather

Background Over Central China Based on Aircraft Observations, Journal of Geophysical Research:

Atmospheres, 125, e2019JD031915, https://doi.org/10.1029/2019JD031915, 2020.

Zhang, L., Li, J., Jiang, Z., Dong, Y., Ying, T., and Zhang, Z.: Clear-Sky Direct Aerosol Radiative Forcing

Uncertainty Associated with Aerosol Optical Properties Based on CMIP6 Models, Journal of Climate,

35, 3007-3019, https://doi.org/10.1175/JCLI-D-21-0479.1, 2022.

Zhang, Q., Ma, X., Tie, X., Huang, M., and Zhao, C.: Vertical distributions of aerosols under different

weather conditions: Analysis of in-situ aircraft measurements in Beijing, China, Atmospheric

Environment, 43, 5526-5535, https://doi.org/10.1016/j.atmosenv.2009.05.037, 2009.







**Table 1.** The aerosols parameters of the look-up table. $m_r$ denotes the real part of the refractive index, $m_i$
denotes the imaginary part of the refractive index, and $s_d$ is the standard deviation of the lognormal size
distribution.

| | Type 1 | Type 2 | Type 3 | Type 4 | Type 5 | Type 6 |
|---|---|---|---|---|---|---|
| $m_r$ (532nm) | 1.414 | 1.517 | 1.380 | 1.404 | 1.400 | 1.452 |
| $m_i$ (532nm) | 0.0036 | 0.0234 | 0.0001 | 0.0063 | 0.0050 | 0.0109 |
| $m_r$ (1064nm) | 1.495 | 1.541 | 1.380 | 1.439 | 1.400 | 1.512 |
| $m_i$ (1064nm) | 0.0043 | 0.0298 | 0.0001 | 0.0073 | 0.0050 | 0.0137 |
| $s_d$ | 1.4813 | 1.5624 | 1.6100 | 1.5257 | 1.6000 | 1.5112 |






**Table 2.** Information of collocated EARLINET and CALIPSO cases.

| Station | Time (UTC) | Horizontal distance (km) |
|---------|------------|--------------------------|
| | 2006-08-20 01:17:25 | 0.0708 |
| | 2007-06-20 01:17:57 | 0.0808 |
| | 2008-07-08 01:18:43 | 0.0690 |
| Napoli | 2008-08-02 01:13:02 | 1.3246 |
| | 2008-08-09 01:19:14 | 0.0807 |
| | 2009-09-29 01:21:03 | 0.0778 |
| | 2019-04-05 02:47:48 | 0.0863 |
| Evora | 2020-01-13 02:54:00 | 0.0164 |
| | 2020-03-18 02:55:43 | 0.0009 |
| | 2015-08-15 01:19:14 | < 0.0001 |
| Warsaw | 2020-03-31 01:13:38 | 0.0177 |



**Figure 1.** The Look-up tables for (a) AE-effective radius, (b) AE-lidar ratio at 532nm and (c) AE-lidar ratio

at 1064nm. The AE is calculated using 532nm and 1064nm aerosol extinction coefficients.





$$S_{532nm} = S^0_{532nm}$$
$$S_{1064nm} = S^0_{1064nm}$$

Solving lidar equation

Extinction coefficient:
$\sigma_{532nm}$、 $\sigma_{1064nm}$

Ångström exponent : AE

AE - $S$ Look-up tables
AE - $\bar{r}$ Look-up tables

Lidar Ratio: $S'_{532nm}$、 $S'_{1064nm}$

Ångström exponent : AE'

Effective radius: $\bar{r}$

Multiple iterations

$$S_{532nm} = S'_{532nm}$$
$$S_{1064nm} = S'_{1064nm}$$

Extinction coefficient:
$\sigma_{532nm}$、 $\sigma_{1064nm}$

Ångström exponent : AE

AE=AE'?

No

Lidar Ratio: $S'_{532nm}$、 $S'_{1064nm}$

Ångström exponent : AE'

AE - $S$ Look-up tables

AE - $\bar{r}$ Look-up tables    Yes

Output: $\bar{r}$、 $\sigma_{532nm}$、 $\sigma_{1064nm}$


**Figure 2.** Schematic of the inversion algorithm ($\lambda_1$ and $\lambda_2$ represent the two different wavelengths,

respectively; S is the lidar ratio; σ is the aerosol extinction; AE is the Ångström index; $\bar{r}$ is the average

particle effective radius; $S^0$ is the initial value of lidar ratio; $S'$ and AE' are the look up values of lidar ratio

and Ångström index, respectively.)





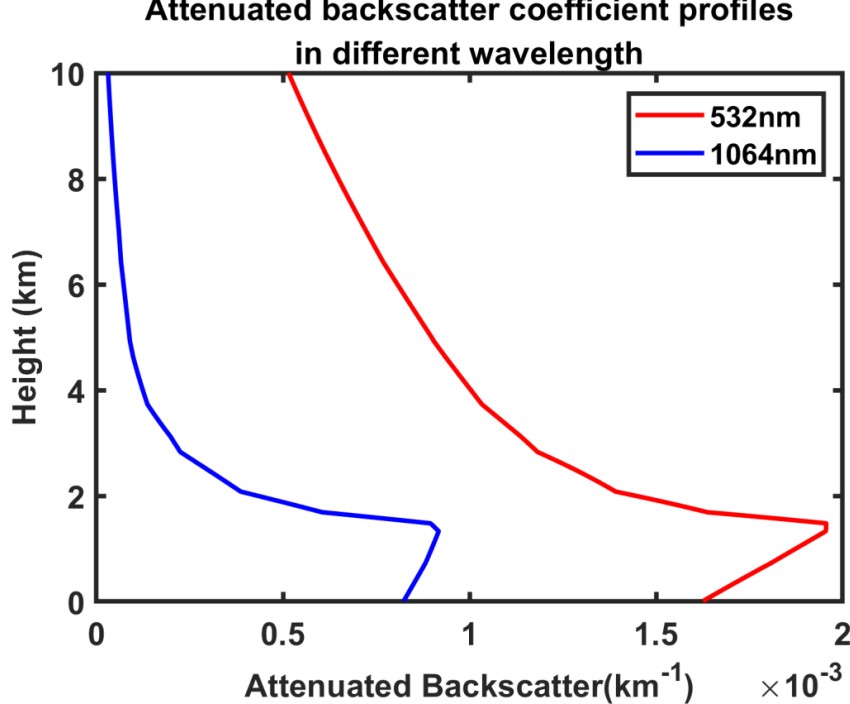


**Figure 3.** The attenuated backscatter coefficient profiles at different wavelengths using syntheticdata.





**Figure 4.** The result of the inversion algorithm using the synthetic data shown in Figure 3.



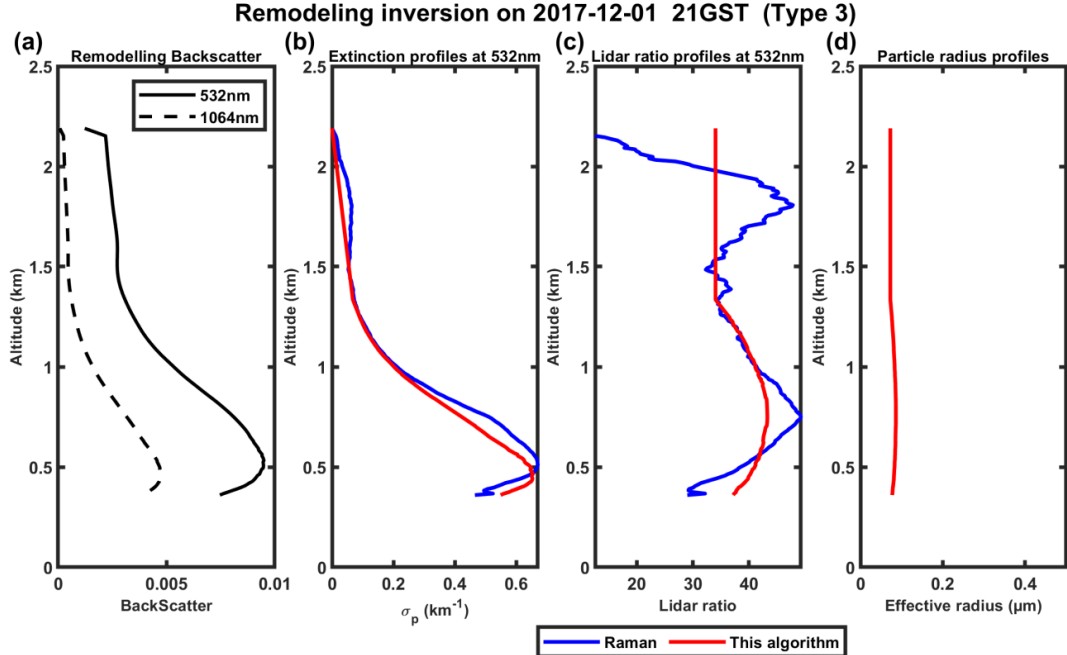


**Figure 5.** (a) Remodeled downward attenuated backscatter profiles measured by Raman lidar in PKU on 1

December 2017; (b) show the extinction profiles inversed by the modified inversion algorithm (red) and

Raman (blue); (c) shows the particle effective radius profiles.






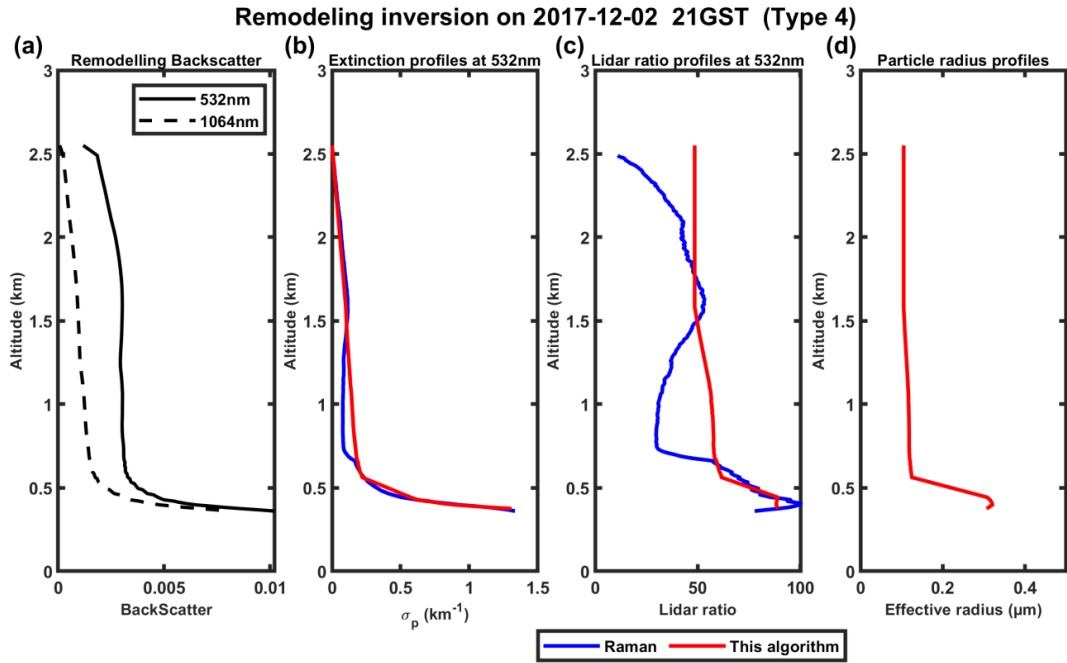


**Figure 6.** Same as Figure 5 but on 2 December 2017.






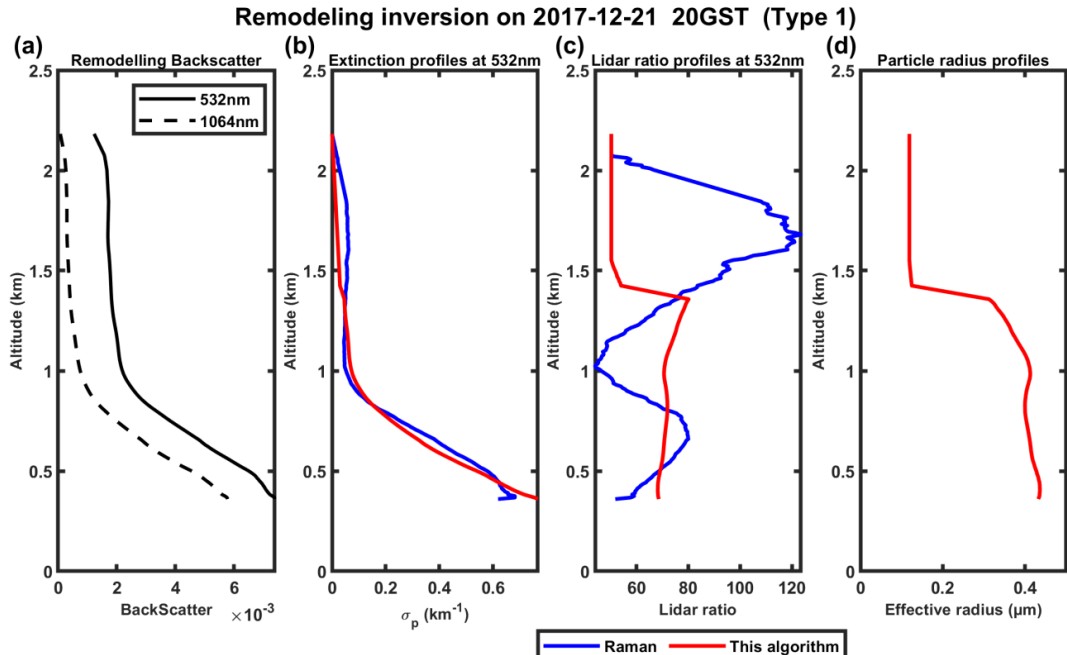

**Figure 7.** Same as Figure 5 but on 21 December 2017.





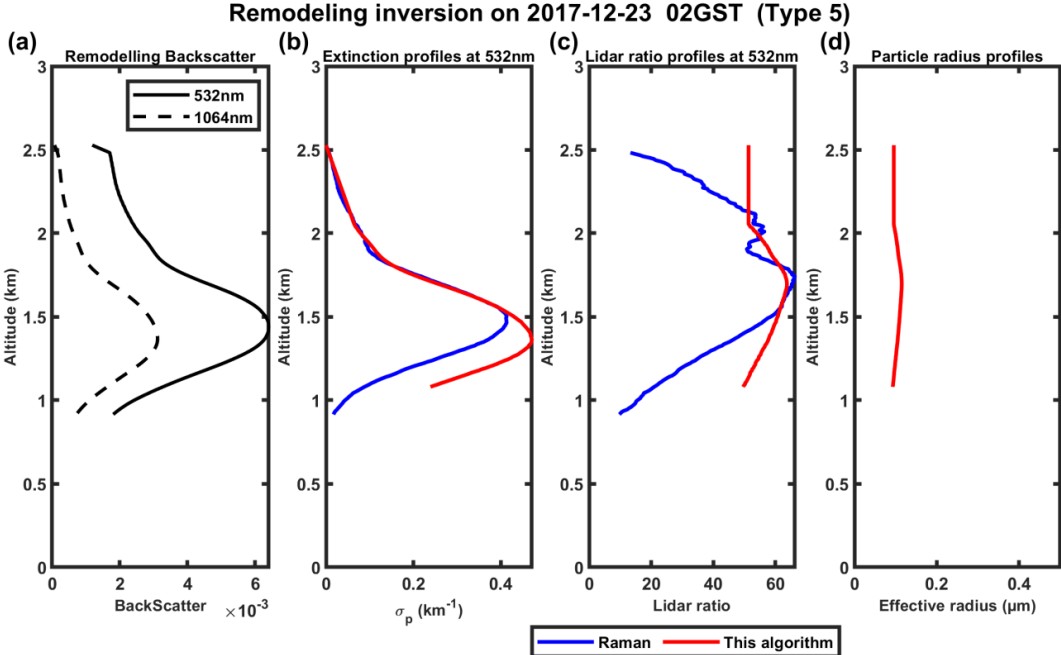


**Figure 8.** Same as Figure 5 but on 23 December 2017.


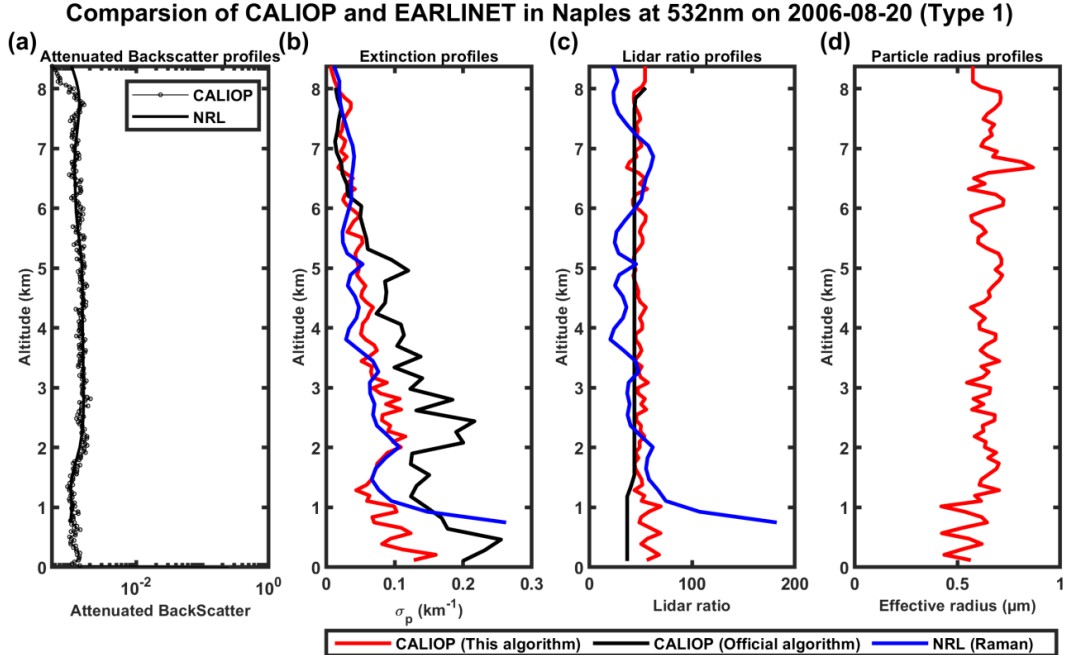

**Figure 9.** 532nm and 106 nm attenuated backscatter profiles measured by CALIOP (black solid line with

circle marker) and NRL (remodeling, black solid line) on 20 August 2006 in logarithmic scale in horizontal

direction (a); (b, c, d) show the extinction profiles, lidar ratio profiles and particle radius profiles,

respectively, provided by our inversion algorithm (red), CALIOP operational level 2 product (black) and

EARLINET level 2 product (blue).





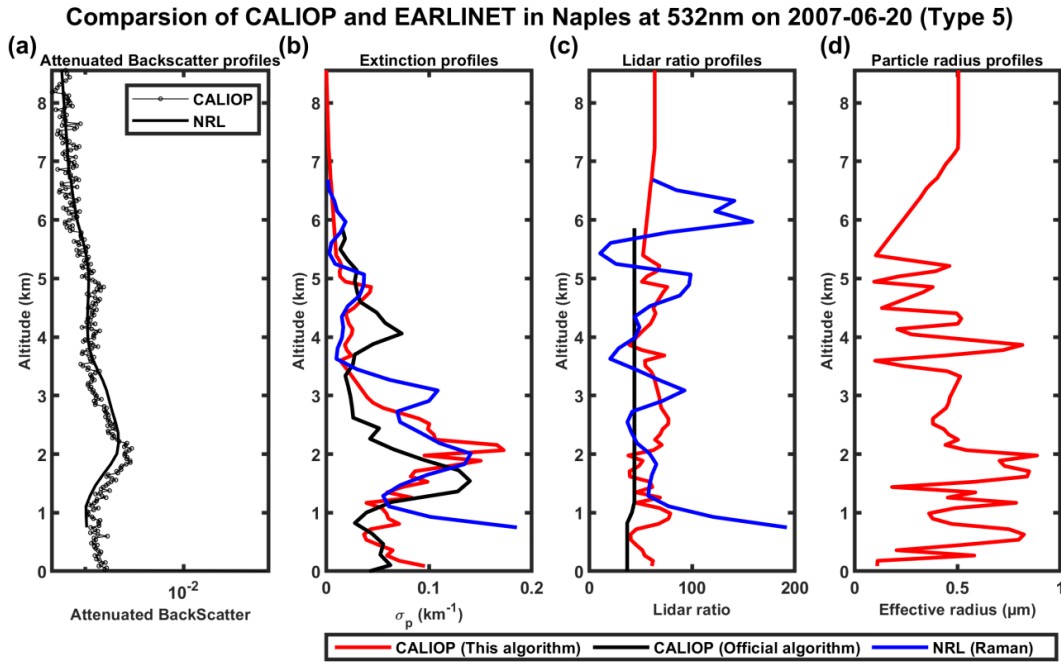


**Figure 10.** Same as Figure 9 but on 20 June 2007.





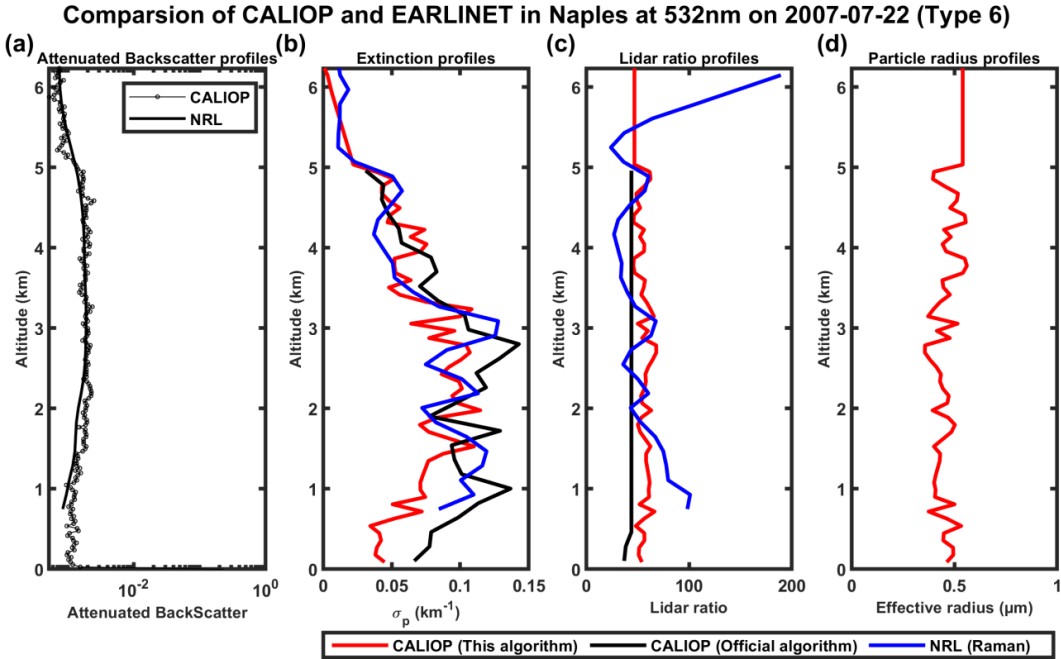


**Figure 11.** Same as Figure 9 but on 22 July 2007.






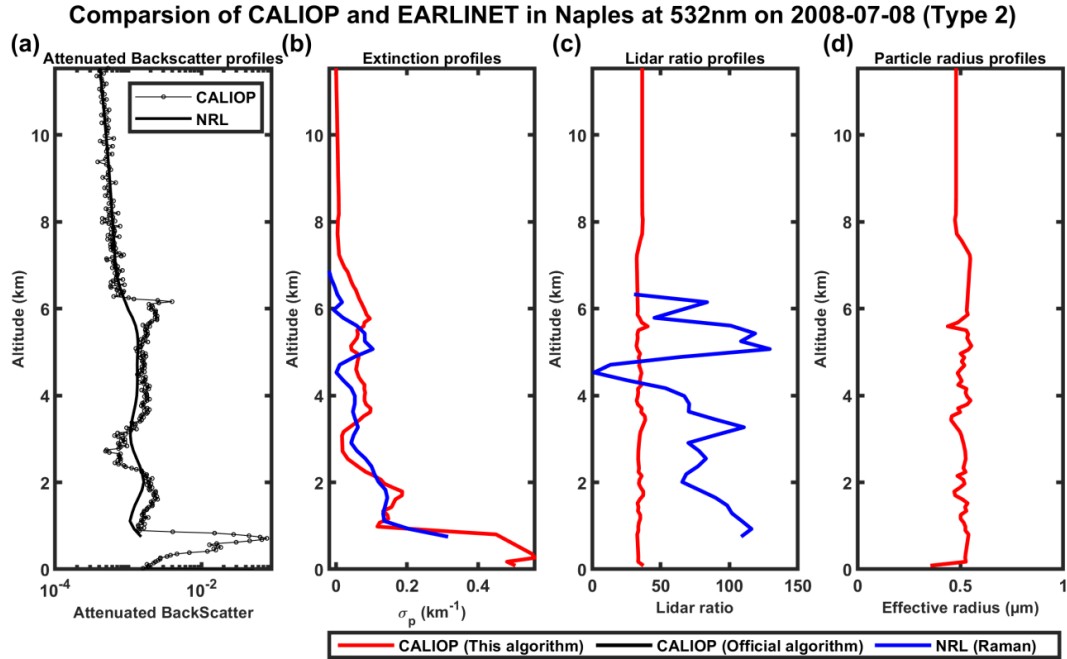


**Figure 12.** Same as Figure 9 but on 8 July 2008.






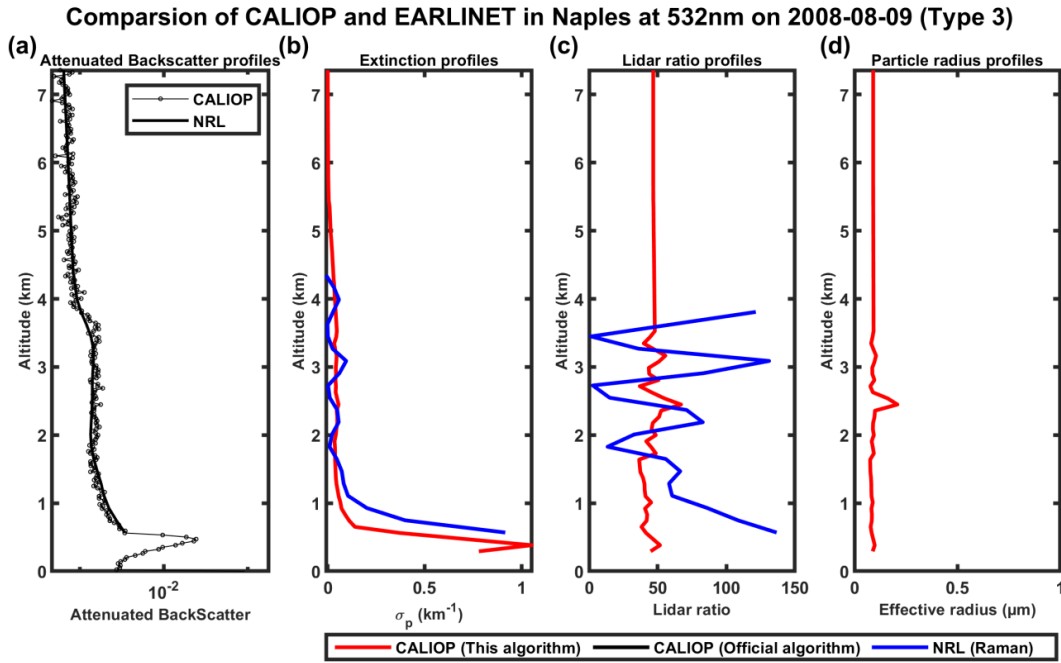


**Figure 13.** Same as Figure 9 but on 9 August 2008.






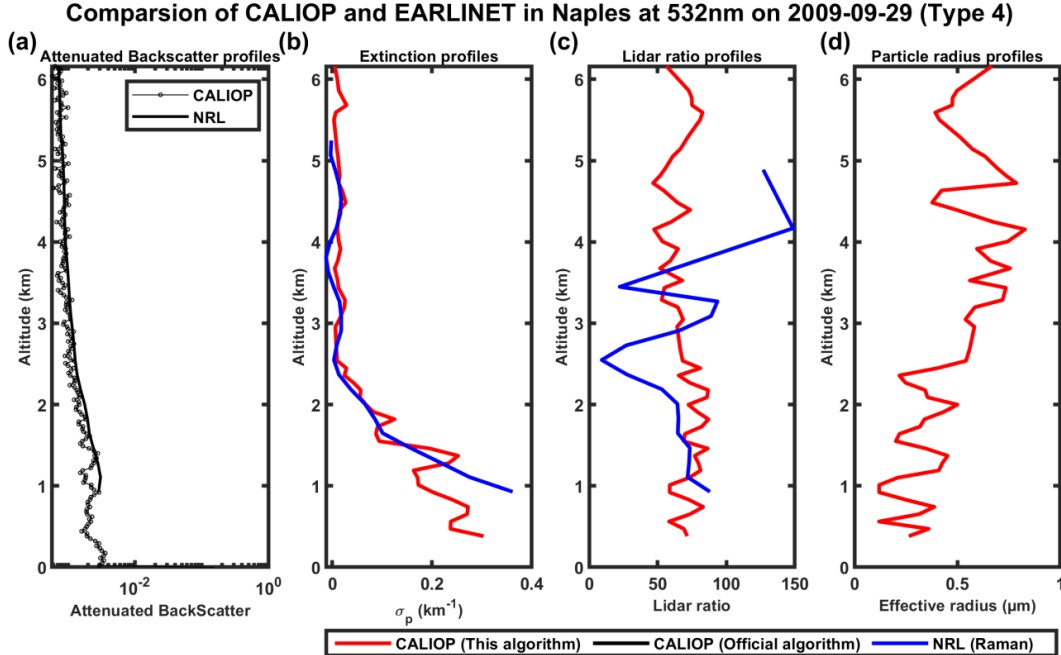


**Figure 14.** Same as Figure 9 but on 29 September 2009.




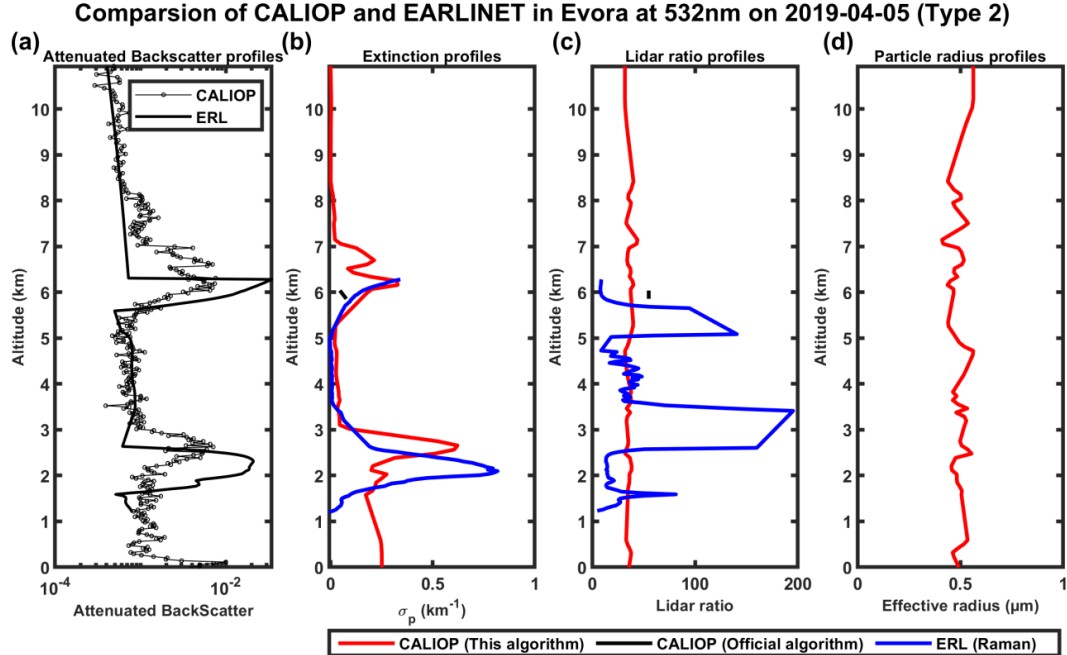


**Figure 15.** 532nm and 106 nm attenuated backscatter profiles measured by CALIOP (black solid line with

circle marker) and ERL at the Evora station (remodeling, black solid line) on 20 August 2006 in

logarithmic scale in horizontal direction (a); (b, c, d) show the extinction profiles, lidar ratio profiles and

particle radius profiles, respectively, provided by the modified inversion algorithm (red), CALIOP level 2

(black) and EARLINET level 2 (blue).





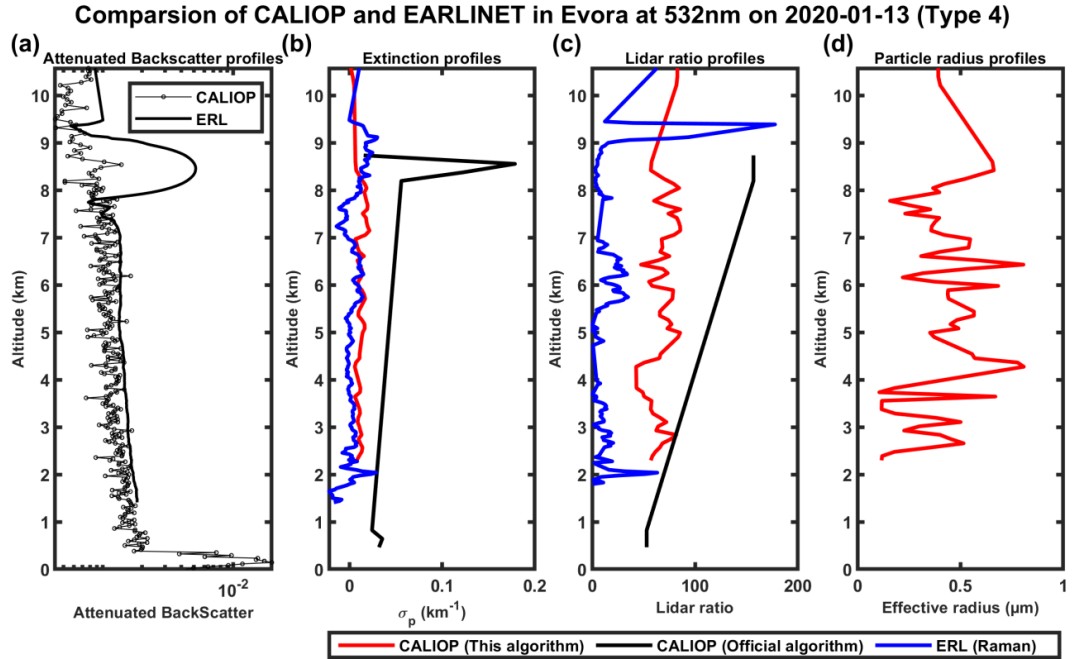

**Figure 16.** Same as Figure 15 but on 13 January 2020.





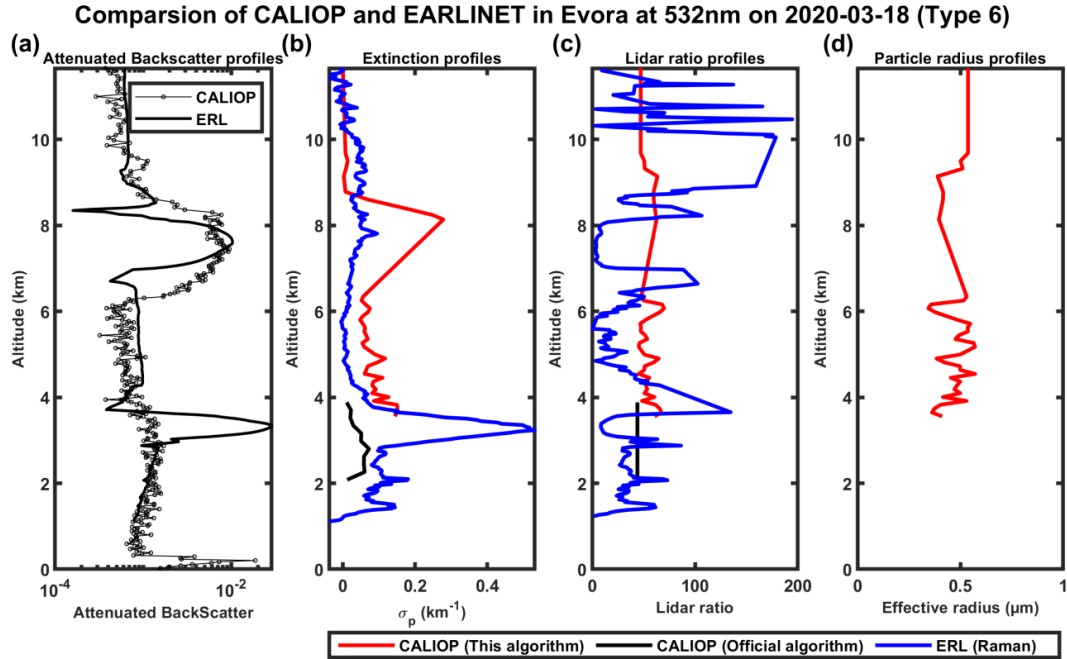

456

**Figure 17.** Same as Figure 15 but on 18 March 2020.

458

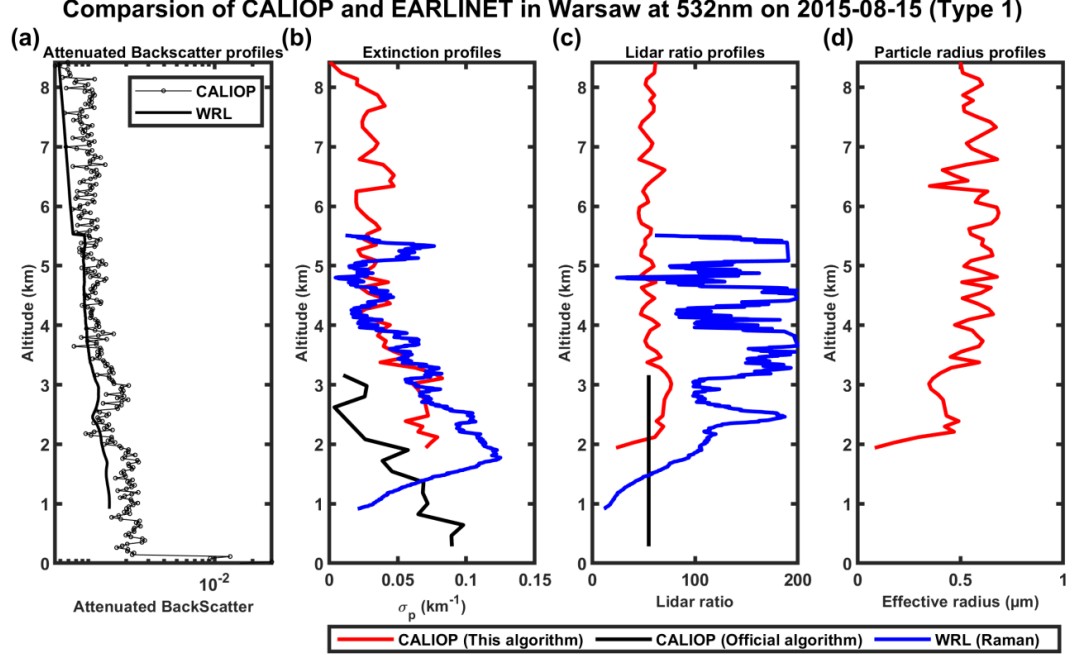

**Figure 18.** 532nm and 106 nm attenuated backscatter profiles measured by CALIOP (black solid line with

circle marker) and WRL at the Warsaw station (remodeling, black solid line) on 20 August 2006 in

logarithmic scale in horizontal direction (a); (b, c, d) show the extinction profiles, lidar ratio profiles and

particle radius profiles, respectively, provided by the modified inversion algorithm (red), CALIOP level 2

(black) and EARLINET level 2 (blue).





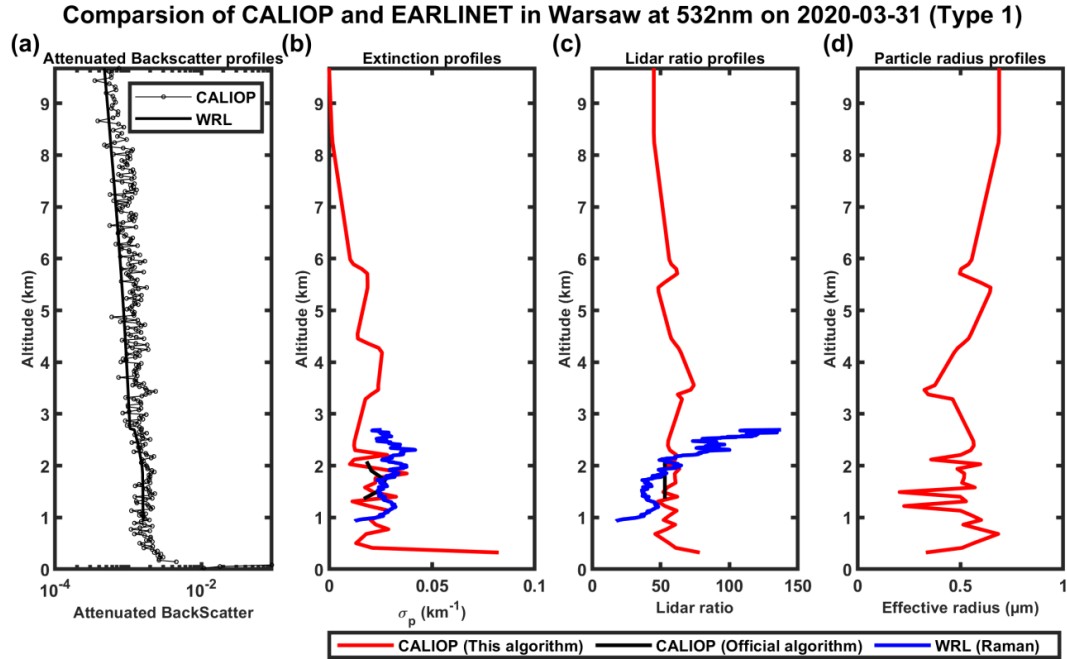


**Figure 19.** Same as Figure 18 but on 31 March 2020.





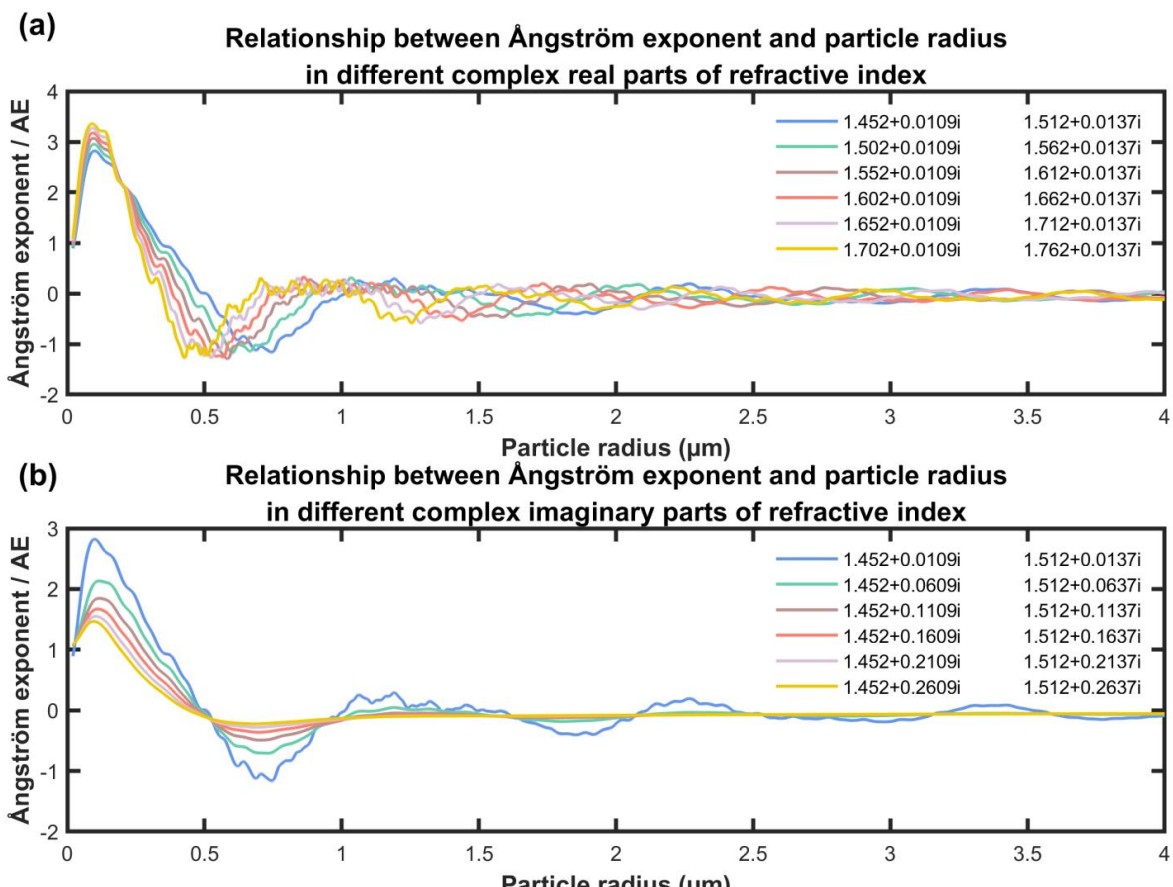


**Figure 20.** The effect of the complex refractive index on Ångström exponent.
