# Peer review of "An iterative algorithm to simultaneously retrieve aerosol extinction and effective radius profiles using"

_Atmospheric Measurement Techniques, 2023_

## Author Comment (AC1)

This study developed a new iterative algorithm to retrieve aerosol extinction and effective radius profile from two-wavelength Mie scattering lidars. The method is justified using synthetic data and applied on CALIPSO measurements. Comparison with EARLINET Raman lidar results indicated improved performance. I recommend publication after addressing the following comments and questions.

Response: We thank the reviewer for his\her encouraging and valuable comments on our manuscript, and we have revised the manuscript according to his/her suggestion.

1. The authors only used Raman lidar measurements from 3 stations to validate the results. Why weren't more sites used?

Response: Thanks for this comment! We did try to collocate all the EARLINET data with CALIOP measurements, but only these three stations are best matched with CALIOP under the collocation and data quality critieria (e.g., clear sky).

2. The comparison between CALIPSO and Raman lidar profiles is too qualitative. Please add some quantitative evaluation. For example, how much is accuracy of extinction profiles improved by the new algorithm?

Response: Thanks for your advice! We have added quantitative evaluation with MAPE (Mean Absolute Percentage Error) in Lines 234-236 as following:

", *and our algorithm reduces the mean MAPE between the retrieval of extinction profiles in CALIOP and Raman lidar from 74% (CALIOP operational product) to 37%.*"

3.    Why do you need to "remodel" Raman lidar profiles?

Response: We are sorry for the confusin. Because CALIOP is a space borne lidar, we use forward integration to solve the lidar equation. In the application to ground-based Raman liadr measurements, because the boundary value of aerosol extinction at surface is very difficult to obtain, we remodel the original lidar signal to the downward attenuated backscatter by lidar equation with molecular & aerosol extinction coefficient and backscatter coefficient profiles obtained from Raman method (Tao et al., 2008), so that we can use the far end solution.

4.    Lines 91-92: The depression "$T2(R)$ is the one-way transmittance from the lidar to the scattering volume at range $R$" may be a mistake. $T2(R)$ in lidar equation is the two-way transmittance.

Response: Thanks for this comment! We have revised the mistake accordingly.

5.    Lines 98: The reference may be cited in a wrong format.

Response: Thanks for your advice! We have revised this reference.

6.    Lines 179-183: How do you obtain the extinction and backscatter coefficient at $1064nm$ from Raman lidar? And what does the approximation of $AE$ at 1064nm meaning?

Response: Thanks for this comment! The Raman inversion just can retrive extinction profiles at 355 nm and 532 nm, as well as backscatter profiles at 355 nm, 532

nm and 1064 nm. We approximate the *532 ~ 1064 nm AE as the the 355 ~ 532 nm AE* , and use Eq. (10) to calculate extinction profile at 1064 nm. We added the following explanation in Lines 188-189:

"*, that we approximate 532 ~ 1064 nm AE as the the 355 ~ 532 nm AE and calculate extinction profile at 1064 nm according to Eq. (10).*"

7. Lines 214-219: The observed atmospheric profile is used to calculate backscatter and extinction coefficient profiles of air molecules for CALIOP retrieval. But where does the observation data come from?

Response: Thanks for this comment! The CALIOP level 1B products include meterological data provided by MERRA-2.

References

Tao, Z., McCormick, M. P., and Wu, D.: A comparison method for spaceborne and
ground-based lidar and its application to the CALIPSO lidar, Applied Physics
B, 91, 639, 10.1007/s00340-008-3043-1, 2008.

---

## Author Comment (AC2)

In this work, the extinction profile and the effective radius are retrieved simultaneously from the CALIOP lidar based on look-up tables. In general, the topic is important and falls within the scope of AMT. However, some revisions should be made before it is considered for publication.

Response: We thank the reviewer for such positive comments on our work! Below we answer the omments point-by-point, and we have also revised the manuscript accordingly.

1. The look-up tables are based on Mie scattering, which assumes spherical aerosols. However, aerosols in the atmosphere have a complex morphology. I do not assume that the morphologies significantly affect the extinction calculations, but you need to point this out. However, when calculating the backscatter factor or backscatter cross section, morphology is expected to have a large influence. Since the lidar properties, AE and polarimetric properties of aerosols are significantly affected by particle shape (Luo et al. 2019, Kahnert et al. 2020, Gialitaki et al.2020, Luo et al. 2022), the authors may need to add some clarifications. We do not expect a large influence on the extinction retrieval, but the effective radius retrieval may be significantly influenced by aerosol shape.

Luo, J., zhang, Q., Luo, J., Liu, J., Huo, Y., & Zhang, Y. (2019). Optical modeling of black carbon with different coating materials: The effect of coating configurations. Journal of Geophysical Research: Atmospheres, 124, 13230–13253.

https://doi.org/10.1029/2019JD031701

Kahnert, M., Kanngießer, F., Järvinen, E., Schnaiter, M. (2020). Aerosol-optics model for the backscatter depolarisation ratio of mineral dust particles. Journal of Quantitative Spectroscopy and Radiative Transfer 254, 107177.

Gialitaki, A., Tsekeri, A., Amiridis, V., Ceolato, R., Paulien, L., Kampouri, A., Gkikas, A., Solomos, S., Marinou, E., Haarig, M., Baars, H., Ansmann, A., Lapyonok, T., Lopatin, A., Dubovik, O., Groß, S., Wirth, M., Tsichla, M., Tsikoudi, I., and Balis, D.: Is the near-spherical shape the "new black" for smoke?, Atmos. Chem. Phys., 20, 14005–14021, https://doi.org/10.5194/acp-20-14005-2020, 2020.

Luo, J., Li, Z., Fan, C., Xu, H., Zhang, Y., Hou, W., Qie, L., Gu, H., Zhu, M., Li, Y., and Li, K.: The polarimetric characteristics of dust with irregular shapes: evaluation of the spheroid model for single particles, Atmos. Meas. Tech., 15, 2767–2789, https://doi.org/10.5194/amt-15-2767-2022, 2022.

Response: Thanks for your advice! We have studied these papers carefully and improved the discussion sections of our manuscript with references in Lines 300-302 as following:

*"possibly by taking advantage of the depolarization ratio (Gialitaki et al., 2020; Kahnert et al., 2020; Luo et al., 2022; Luo et al., 2019) measurement that is not used here."*

2.    Equation 12: The authors should clarify the definition of $Qb(\lambda, r)$ and how to calculate the optical properties (which code?). I think the $Qb(\lambda, r)$ is calculated based on the scattering efficiency and phase function at the backward angles. Please clarify it.

Response: We are sorry for the confusion. $Q_b(\lambda, r)$ is the scattering efficiency of the particle at 180° calculated with phase function. We use the Lorenz–Mie scattering FORTRAN program (Mishchenko and Yang, 2018) to obtain the optical properties ($Q_e(\lambda, r)$ and $Q_b(\lambda, r)$). We have revised the manuscript accordingly in Lines 128-131 as following:

"$Q_e(\lambda, r)$ and $Q_b(\lambda, r)$ denote the extinction and backscatter efficiencies of the particle (the scatter factor of the particle at 180°) with size $r$ at wavelength $\lambda$ respectively. The size parameter is defined as $x \equiv 2\pi r / \lambda$, where $1 < x < 50$ for typical aerosols and thus the Mie scattering theory (Mishchenko and Yang, 2018) can be applied."

3.    Please clarify how to retrieve the optical properties, which object function?

Response: Thanks for this comment!    As shown in Figure 2, we retrieve the optical properties by solving the lidar equation using the Fernald method (Fernald, 1984) by establishing a look-up table. Firstly, we calculate the extinction coefficients ($\sigma_{532\ nm}$ & $\sigma_{1064\ nm}$) of two wavelengths ($532\ nm$ & $1064\ nm$) from an initial guess of the lidar ratios ($S^0_{532\ nm}$ & $S^0_{1064\ nm}$) by solving the lidar equation (Eq. 6), and then obtain the Ångström exponent ($AE$) through Eq. (10). Secondly, the look-up table is used to

determine a set of new lidar ratios ($S'_{532\,nm}$ & $S'_{1064\,nm}$), which is used to calculate the

new $\sigma_{532\,nm}$ & $\sigma_{1064\,nm}$ and Ångström exponent ($AE'$). This procedure is repeated

until the difference between the updated $AE'$ and previous $AE$ reduces to a very small

value ($10^{-3}$). The final values of $\sigma_{532\,nm}$, $\sigma_{1064\,nm}$, $S_{532\,nm}$ and $S_{1064\,nm}$ are the

retrieved optical properties of this layer, and the backscatter coefficients $\beta_{532\,nm}$ and

$\beta_{1064\,nm}$ can also be obtained by Eq. (5). We have revised the manuscript accordingly

in Lines 77-78 and Lines 156-157 as following:

*"by solving the lidar equation using the Fernald method (Fernald, 1984) with a*

*look-up table approach in the iteration procedure."*

*"and the final values of $\sigma_{532\,nm}$, $\sigma_{1064\,nm}$, $S_{532\,nm}$, $S_{1064\,nm}$ and $\bar{r}$ are the*

*retrieved results of this layer."*

4.      Equation 13: I think that this equation does not reflect the lognormal distribution,

and this is the normal distribution. Please modify it. Is "the average particle radius" "the

geometric mean radius"? is   "standard deviation" "geometric standard deviation"?

Response: We are sorry for this mistakes. The "standard deviation" is "geometric

standard deviation", and "the average particle radius" actually is "the median radius".

We have revised the manuscript accordingly in Lines 132-139 as following:

*"As the limited information provided by two-wavelength lidar, we assume the*

*volume-size distribution of aerosols conform to the lognormal distribution, and the size*

*distribution is expressed as follows (Deshler et al., 2003; Hara et al., 2021):*

$$n(r) = \frac{N}{r \ln s_d \sqrt{2\pi}} e^{-\frac{(\ln r - \ln r_0)^2}{2(\ln s_d)^2}}, \qquad\qquad (13)$$

*Where N is the total particle concentrations; $r_0$ and $s_d$ are the median radius and the geometric standard deviation of aerosol size distribution, respectively. When we assumed a constant $s_d$ for the same aerosol, the AE can be calculated when given an $r_0$. The particle effective radius ($\bar{r}$) is defined by:*

$$\bar{r} = \frac{\sum n(r)r^3}{\sum n(r)r^2}, \qquad\qquad (14)"$$

5.    Line 137 "When we assumed a constant $sd$ for the same aerosol""the AE can be calculated when given an r" ?

Response: Thanks for your advice! We have revised the sentence accordingly in Lines 137-138 as following:

"*When we assumed a constant $s_d$ for the same aerosol type, the AE can be calculated with a given $r_0$ value.*"

6.    Line 138-139: Please provide references.

Response: Thanks for your advice! We have added the references following this sentence in Lines 140-141 as:

"*We choose the six types of aerosols with their parameters in Table 1, which is consistent with the aerosol classification used in the operational algorithm of CALIOP (Winker et al., 2009).*"

7.    Lines 116-117. I do not understand why it says: "The two-wavelength lidar can give two independent profiles of the attenuated backscatter coefficients", and why the

profiles of the aerosol extinction coefficients were calculated based on the attenuated backscatter coefficients. From equation 6, the aerosol extinction coefficient profiles can be determined. This theorem is confusing.

Response: We are sorry for the confusion. The attenuated backscatter coeffcients profiles, provided by the measurements of CALIOP with calibrated and range-corrected, are the source data for our inversion algorithm. $E(R)$ in Eq. (6) is calculated by Eq. (4) with the attenuated backscatter coeffcients $\beta'(R)$ and the transmittance of air molecules $T_m^2(R)$, which means that the profiles of the aerosol extinction coefficients must be calculated based on the attenuated backscatter coefficients. The two-wavelength lidar have two independent measurements of attenuated backscatter coefficients. We have have revised this sentence in Lines 114-115 as following:

"*The two-wavelength lidar can give two independent profiles of attenuated backscatter coefficients at different wavelengths,*"

8. How do you define the effective radius? Please give a definition. Have you used the geometric mean as a substitute for the effective radius? Please explain this.

Response: Thanks for your advice! We have revised the manuscript accordingly in Lines 138-139 as following:

"*The particle effective radius ($\bar{r}$) is defined by:*

$$\bar{r} = \frac{\sum n(r)r^3}{\sum n(r)r^2}, \qquad\qquad (14)"$$

9. The authors say that the CALIOP products classify the aerosols into different

types and that this is not sufficient to represent the aerosols in the atmosphere, but the refractive indices and the geometric standard deviation of the size distributions from CALIOP are still used and only change the geometric mean radius. Please justify the use of these parameters, as you have said that this is not sufficient.

Response: Thanks for this comment! Indeed the CALIOP classification might be insufficient, especially in our application we find that some profiles could not yield an valid retrieval, which is very likely duet to limitations of the look-up table constructed with CALIOP aerosol type. However, the CALIOP types are contructed using AERONET observations, which is the most extensive ground based aerosol network up to date. It is therefore very difficult to overcome the limitations in the aerosol types for the moment. We also plan to refine our look-up table to improve the retrieval accuracy in the future once there are more surface observations. We have thus removed "*However, due to the limited coverage and spatial representativeness of surface stations, these lidar ratio assumptions may not be appropriate or representative at certain locations (Josset et al., 2011), which is an important source of retrieval uncertainty.*" from the manuscript.

References

Deshler, T., Hervig, M. E., Hofmann, D. J., Rosen, J. M., and Liley, J. B.: Thirty years of in situ stratospheric aerosol size distribution measurements from Laramie,

Wyoming (41°N), using balloon-borne instruments, Journal of Geophysical Research: Atmospheres, 108, 10.1029/2002jd002514, 2003.

Fernald, F. G.: Analysis of atmospheric lidar observations: some comments, Appl. Opt., 23, 652-653, 10.1364/AO.23.000652, 1984.

Gialitaki, A., Tsekeri, A., Amiridis, V., Ceolato, R., Paulien, L., Kampouri, A., Gkikas, A., Solomos, S., Marinou, E., Haarig, M., Baars, H., Ansmann, A., Lapyonok, T., Lopatin, A., Dubovik, O., Groß, S., Wirth, M., Tsichla, M., Tsikoudi, I., and Balis, D.: Is the near-spherical shape the "new black" for smoke?, Atmos. Chem. Phys., 20, 14005-14021, 10.5194/acp-20-14005-2020, 2020.

Hara, K., Nishita-Hara, C., Osada, K., Yabuki, M., and Yamanouchi, T.: Characterization of aerosol number size distributions and their effect on cloud properties at Syowa Station, Antarctica, Atmos. Chem. Phys., 21, 12155-12172, 10.5194/acp-21-12155-2021, 2021.

Kahnert, M., Kanngießer, F., Järvinen, E., and Schnaiter, M.: Aerosol-optics model for the backscatter depolarisation ratio of mineral dust particles, Journal of Quantitative Spectroscopy and Radiative Transfer, 254, 107177, https://doi.org/10.1016/j.jqsrt.2020.107177, 2020.

Luo, J., Zhang, Q., Luo, J., Liu, J., Huo, Y., and Zhang, Y.: Optical Modeling of Black Carbon With Different Coating Materials: The Effect of Coating Configurations, Journal of Geophysical Research: Atmospheres, 124, 13230-13253, https://doi.org/10.1029/2019JD031701, 2019.

Luo, J., Li, Z., Fan, C., Xu, H., Zhang, Y., Hou, W., Qie, L., Gu, H., Zhu, M., Li, Y., and Li, K.: The polarimetric characteristics of dust with irregular shapes: evaluation of the spheroid model for single particles, Atmos. Meas. Tech., 15, 2767-2789, 10.5194/amt-15-2767-2022, 2022.

Mishchenko, M. I. and Yang, P.: Far-field Lorenz–Mie scattering in an absorbing host medium: Theoretical formalism and FORTRAN program, Journal of Quantitative Spectroscopy and Radiative Transfer, 205, 241-252, https://doi.org/10.1016/j.jqsrt.2017.10.014, 2018.

Winker, D. M., Vaughan, M. A., Omar, A., Hu, Y., Powell, K. A., Liu, Z., Hunt, W. H., and Young, S. A.: Overview of the CALIPSO Mission and CALIOP Data Processing Algorithms, Journal of Atmospheric and Oceanic Technology, 26, 2310-2323, 10.1175/2009jtecha1281.1, 2009.

---

## Author Comment (AC3)

Review of "An iterative algorithm to simultaneously retrieve aerosol extinction and effective radius profiles using the CALIOP lidar" by Liang Chang et al.

Recommendation: Minor Revisions

The manuscript "An iterative algorithm to simultaneously retrieve aerosol extinction and effective radius profiles using the CALIOP lidar" provides a modified two-wavelength lidar inversion algorithm to retrieve the vertical distribution of both aerosol extinction and particle effective radius. The study built a look-up table to relate the lidar ratio with the Ångström exponent calculated using aerosol extinction at the two wavelengths. In order to verify the accuracy of the algorithm, two different lidar data (ground-based Raman lidar and CALIOP) were used for the application and the results showed good agreement.

In general, the paper presented in a logical way, but lightly lacking in English expression. The algorithm has some prospects of practical application. I therefore recommend publication of this paper in Atmospheric Measurement Techniques after minor revisions. My comments are listed as follows:

Response: We thank the reviewer very much for his/her encouraging comments, and we have revised the manuscript according to these specific comments and technical corrections.

Specific Comments:

1. Table 1 shows the aerosols parameters of the look-up table. How were these parameters obtained? If derived from experimental or simulation results, please

explain in the manuscript and provide references.

Response: Sorry for the confusion. These parameters are obtained from aerosol classification used in the operational algorithm of CALIOP, and we have added the references following this sentence.

2. In line 148 of the manuscript, it is mentioned "an initial guess". How did this initial guess work out? Does the selection of values for the initial guess affect the final results obtained.

Response: We guess the the initial value from the Look-up tables (Figure 1). For example, when AE is minimum, the lidar ratios are about 40 at 532 nm and 60 at 1064 nm. In the test of the inversion alagorithm with synthetic data, we test several sets of initial lidar ratios (e.g., 60 at 532 nm & 80 at 1064 nm, 80 at 532 nm & 60 at 1064 nm, 40 at 532 nm & 30 at 1064 nm, etc.), and find that using these initial value can all reach similar retrieval results. The only difference is the number of iterations. Therefore, the initial lidar ratios of 40 at 532 nm and 60 at 1064 nm are used in the following retrieval.

3. In line 18 How are these four types of aerosols distinguished and determined? Please explain in the manuscript.

Response: We are sorry for the confusion. On the test of the inversion alagorithm with synthetic data, we find that selection of aerosol type is critical as incorrect assumption of aerosol refractive index will result in divergence of the algorithm and fail to yield valid retrieval. Similar behavior is noted in the application to real lidar

measurements. Thus, we determine the aerosol type by selecting the one that yields the best retrieval results. We added the following explanation in Lines 177-178:

"*This also helps us to determine the appropriate aerosol type, i.e., the type that yields the best retrieval results.*"

4. Figure 5(d) shows the results of the aerosol effective radius profiles obtained from the inversion, but without corresponding comparative validation results. How can the accuracy of the algorithm be demonstrated?

Response: We are sorry that we couldn't find the true values of the aerosol effective radius profiles to validate our retrieved results. As a result, we can only infer the validity of the retrieval empirically according to published results (Liu et al., 2009; Zhang et al., 2009; Yang et al., 2020; Cai et al., 2022; Li et al., 2022) and physical theory.

5. In Section 4, uncertainty analysis, the authors have only made a general analysis without giving specific values; various assumptions are used in the algorithm and the corresponding uncertainty analysis should be given.

Response: Thanks for your advice! We expanded the discussion about uncertainties associated assumptions, added more quantitative results in the uncertainty analysis, and used the MAPE (Mean Absolute Percentage Error) to quantify the error. The revised text in Lines 260-279 is cited as follows:

"*Uncertainties in aerosol extinction and effective radius profiles retrieved by our two-*

*wavelength inversion algorithm are mainly due to measurement noise (e.g., the signal statistical error, the estimations of molecular optical properties, etc.), calibration errors, and assumption errors. In this section, we further examine the errors associated with the assumptions in the algorithm.*

*First, the single-scattering approximation is used in solving lidar equation, as multiple scattering effects in aerosol layers are generally small and are currently neglected for CALIOP (Winker et al., 2009). We limit the application of our algorithm to clear sky weather conditions to reduce this error, but this error is very difficult to quantify.*

*Second, the errors in the aerosol refractive index, size distribution and spherity assumptions in look-up tables can also introduce errors in solving the lidar equation. The lognormal distribution assumption of aerosol volume-size distribution may make the algorithm fail to converge in other actual size distributions. For example, using data generated by Junge distribution (a simpler aerosol size distribution), the algorithm cannot yield valid retrieval results. Similar outcome is noted for non-spherical particles or aerosol types significantly different from the assumed type.*

*Finally, we consider assumption and retrieval unceratinties as a perturbation in the lidar ratio and attempt to quantify its effect on the retrieved profiles. We increase the lidar ratio profiles at 532 nm and 1064 nm from the look-up tables by ±10% before calculating the synthetic attenuated backscatter profiles, which makes the synthetic data do not entirely match the look-up table. The retrieved profiles exhibit mean MAPE less than 14% (lidar ratio increases by 10%) and 17% (lidar ratio decreases by 10%),*

*indicating that the algorithm is comparatively robust to noise."*

Technical Corrections

1. Line 32: The literature (Ipcc,2023) should be changed to (IPCC, 2023).

    Response: Thanks for your advice! We have revised the manuscript accordingly.

2. Line 43: There should be a space between the number and the unit (532 nm) and line 179 (53 m).

    Response: Thanks for your advice! We have revised all the mistake units in manuscript accordingly.

3. In Section 3.1, the unit "nm" has different fonts, please standardize the format.

    Response: Thanks for your advice! We have revised all the mistake in manuscript accordingly.

4. Line 125: "with some size distribution", the word 'some' doesn't seem to fit here.

    Response: Thanks for your advice! We have revised this sentence as:

    "*By assuming spherical particles size distribution*"

5. Line 166: "To save space" can be removed. should be a space between the number and the unit (1 km).

    Response: Thanks for your advice! We have removed "*To save space*" in the

sentence, and revised all the mistake units in manuscript accordingly.

6. There is only one red line in figure 5(d), so the subtitle can be revised to "Particle radius profile".

Response: Thanks for your advice! We have revised all the related figure titles.

References

Cai, Z., Li, Z., Li, P., Li, J., Sun, H., Yang, Y., Gao, X., Ren, G., Ren, R., and Wei, J.: Vertical distributions of aerosol microphysical and optical properties based on aircraft measurements made over the Loess Plateau in China, Atmospheric Environment, 270, 118888, https://doi.org/10.1016/j.atmosenv.2021.118888, 2022.

Li, Y., Guo, X., Jin, L., Li, P., Sun, H., Zhao, D., and Ma, X.: Aircraft Measurements of Summer Vertical Distributions of Aerosols and Transitions to Cloud Condensation Nuclei and Cloud Droplets in Central Northern China, Chinese Journal of Atmospheric Sciences, 46, 845, 10.3878/j.issn.1006-9895.2104.20255, 2022.

Liu, P., Zhao, C., Zhang, Q., Deng, Z., Huang, M., Ma, X., and Tie, X.: Aircraft study of aerosol vertical distributions over Beijing and their optical properties, Tellus B, 61, 756-767, https://doi.org/10.1111/j.1600-0889.2009.00440.x, 2009.

Winker, D. M., Vaughan, M. A., Omar, A., Hu, Y., Powell, K. A., Liu, Z., Hunt, W. H., and Young, S. A.: Overview of the CALIPSO Mission and CALIOP Data Processing Algorithms, Journal of Atmospheric and Oceanic Technology, 26, 2310-2323, 10.1175/2009jtecha1281.1, 2009.

Yang, J., Li, J., Li, P., Sun, G., Cai, Z., Yang, X., Cui, C., Dong, X., Xi, B., Wan, R., Wang, B., and Zhou, Z.: Spatial Distribution and Impacts of Aerosols on Clouds Under Meiyu Frontal Weather Background Over Central China Based on Aircraft Observations, Journal of Geophysical Research: Atmospheres, 125, e2019JD031915, https://doi.org/10.1029/2019JD031915, 2020.

Zhang, Q., Ma, X., Tie, X., Huang, M., and Zhao, C.: Vertical distributions of aerosols under different weather conditions: Analysis of in-situ aircraft measurements in Beijing, China, Atmospheric Environment, 43, 5526-5535, https://doi.org/10.1016/j.atmosenv.2009.05.037, 2009.

---

## Author Response (AR2)

Many thanks to the author for the careful revision and the significant improvement in the quality of the work. I have only one small question:

It is recommended that the author discuss in more detail why the aerosol parameters of CALIOP are used, only the geometric mean particle size is changed, and why the geometric standard deviation is not changed. If the authors agree with the aerosol type of CALIOP, there is no need to change the geometric mean particle size. If the authors disagree, other parameters should also be changed. Since the aerosol type parameters of CALIOP were calculated statistically, the author should not assume that the other parameters are fixed and change only one particular parameter. Please justify the choice of parameters.

Response: We thank the reviewer very much for his/her encouraging comments and we have revised the manuscript according to the suggestion in Lines 138-143 as following:

"*Where $N$ is the total particle concentrations; $r_0$ and $s_d$ are the median radius and the geometric standard deviation of aerosol size distribution, respectively. The particle size distribution is represented by its effective radius ($\bar{r}$) defined as:*

$$\bar{r} = \frac{\sum n(r)r^3}{\sum n(r)r^2}, \tag{14}$$

*For convenient calculation, we assume a constant $s_d$ for the each aerosol type, and the relationship between AE and $\bar{r}$ can be established with given $r_0$ values.*"

---

## Author Response (AR3)

Public justification (visible to the public if the article is accepted and published):

Please find the minor suggestions by the reviewer.

It is recommended that the author discuss in more detail why the aerosol parameters of CALIOPare used, only the geometric mean particle size is changed, and why the geometric standard deviation is not changed. If the authors agree with the aerosol type of CALIOP, there is no need to change the geometric mean particle size. If the authors disagree, other parameters should also be changed. Since the aerosol type parameters of CALIOP were calculated statistically, the author should not assume that the other parameters are fixed and change only one particular parameter. Please justify the choice of parameters.

Response: We thank the reminder of editor very much and we are sorry for the

confusion. On the one hand, the effective radius is an important parameter indicating particle size distribution, and have been widely considered in the remote sensing of aerosols (Veselovskii et al., 2002; Di Girolamo et al., 2022). On the other hand, the refractive index and geometric standard deviation of the six aerosol types are used in establishing our look-up talbe, which is consistent with the aerosol classification used in the operational algorithm of CALIOP (Winker et al., 2009). $r_0$ is the median radius of aerosol size distribution in Eq.13, while the geometric mean particle size is not concerned in our manuscript. We have added the references in Line 140, and modified the effective radius from $\bar{r}$ to $r_e$ in manuscript accoedingly.

**References**

Di Girolamo, P., De Rosa, B., Summa, D., Franco, N., and Veselovskii, I.: Measurements of Aerosol Size and Microphysical Properties: A Comparison Between Raman Lidar and Airborne Sensors, Journal of Geophysical Research: Atmospheres, 127, e2021JD036086, https://doi.org/10.1029/2021JD036086, 2022.

Veselovskii, I., Kolgotin, A., Griaznov, V., Müller, D., Wandinger, U., and Whiteman, D. N.: Inversion with regularization for the retrieval of tropospheric aerosol parameters from multiwavelength lidar sounding, Appl. Opt., 41, 3685-3699, 10.1364/AO.41.003685, 2002.

Winker, D. M., Vaughan, M. A., Omar, A., Hu, Y., Powell, K. A., Liu, Z., Hunt, W. H., and Young, S. A.: Overview of the CALIPSO Mission and CALIOP Data

Processing Algorithms, Journal of Atmospheric and Oceanic Technology, 26, 2310-2323, 10.1175/2009jtecha1281.1, 2009.